# Proximal Epithelioid Sarcoma Mimicking Inguinal Inflammation

**DOI:** 10.3390/diagnostics15172246

**Published:** 2025-09-04

**Authors:** Tomonori Kawasaki, Takuya Watanabe, Satoshi Kanno, Tomoaki Torigoe, Kojiro Onohara, Masanori Wako, Tetsuhiro Hagino, Jiro Ichikawa

**Affiliations:** 1Department of Pathology, Saitama Medical University International Medical Center, Hidaka, Saitama 350-1298, Japan; tomo.kawasaki.14@gmail.com (T.K.); kanno820@saitama-med.ac.jp (S.K.); 2Department of Orthopaedic Oncology and Surgery, Saitama Medical University International Medical Center, Hidaka, Saitama 350-1298, Japan; tw52831@5931.saitama-med.ac.jp (T.W.); ttorigoe@saitama-med.ac.jp (T.T.); 3Department of Radiology, Interdisciplinary Graduate School of Medicine, University of Yamanashi, Chuo, Yamanashi 409-3898, Japan; konohara@yamanashi.ac.jp; 4Department of Orthopaedic Surgery, Interdisciplinary Graduate School of Medicine, University of Yamanashi, Chuo, Yamanashi 409-3898, Japan; wako@yamanashi.ac.jp (M.W.); tetsuhiro.hagino@gmail.com (T.H.)

**Keywords:** epithelioid sarcoma, proximal type, inguinal inflammation, imaging, histopathology

## Abstract

Epithelioid sarcoma (ES) is an extremely rare sarcoma categorized into classic and proximal types. Proximal ES is characterized by its occurrence in older individuals, proximal locations, deep tissue involvement, and a tendency to be larger in size. We present a case of an extremely small proximal ES occurring in the inguinal region, which posed significant diagnostic challenges. Clinically, the lesion presented as a painful mass, and magnetic resonance imaging findings suggested lymphadenitis or other inflammatory lesions due to its small size and internal signal patterns. Despite being monitored, the mass showed progression, prompting an incisional biopsy that raised suspicion for ES. Positron emission tomography–computed tomography confirmed the absence of metastases, leading to wide excision. Pathological examination of the excised specimen confirmed proximal ES with negative margins. In this case, characteristic features of proximal ES were scarcely observed, and imaging findings were not distinctive, likely due to the small size of the lesion. Furthermore, the broad differential diagnoses for inguinal masses necessitate careful attention during diagnosis. For sarcomas and tumors in general, reliance solely on clinical and imaging findings can lead to diagnostic pitfalls, emphasizing the importance of active pathological evaluation through biopsy.

**Figure 1 diagnostics-15-02246-f001:**
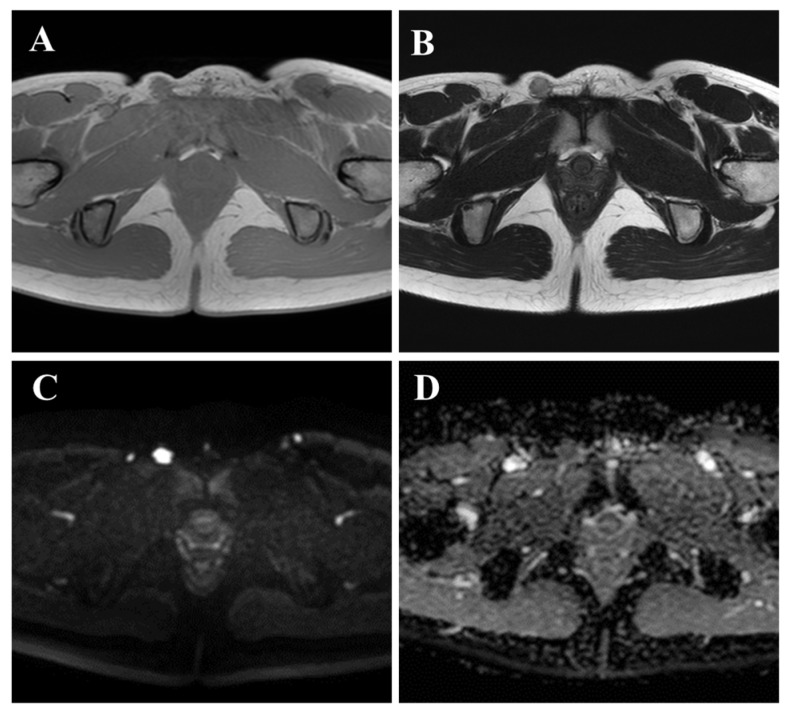
A 21-year-old woman presented with a painful inguinal mass. The mass had first appeared 13 months earlier. Eight months prior, she had consulted a dermatologist. Magnetic resonance imaging (MRI) findings suggested lymphadenitis or an inflammatory pseudotumor, and the lesion was managed conservatively. However, the mass gradually enlarged. At the time of initial presentation, the mass was tender and painful, without redness or local heat. MRI at the previous hospital showed an isointense lesion on T1-weighted images (**A**), faint hyperintensity on T2-weighted images with a capsule-like low-signal rim measuring 14 × 12 × 12 mm (**B**), and diffusion-weighted imaging (DWI) showed high signal intensity (**C**) with low apparent diffusion coefficient (ADC) values (**D**). Although she was monitored without receiving any medications, including antibiotics, due to progressive enlargement and to establish a correct diagnosis, an incisional biopsy was performed, revealing features consistent with epithelioid sarcoma (ES).

**Figure 2 diagnostics-15-02246-f002:**
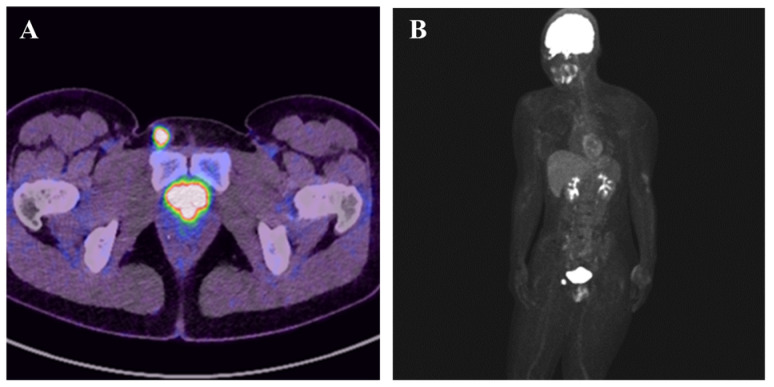
F-fluorodeoxyglucose (FDG)-Positron emission tomography–computed tomography (PET-CT) showed high uptake in the right inguinal region, with a maximum standardized uptake value of 21.2, and no other regions of uptake were noted (**A**,**B**). Therefore, a wide local excision was performed.

**Figure 3 diagnostics-15-02246-f003:**
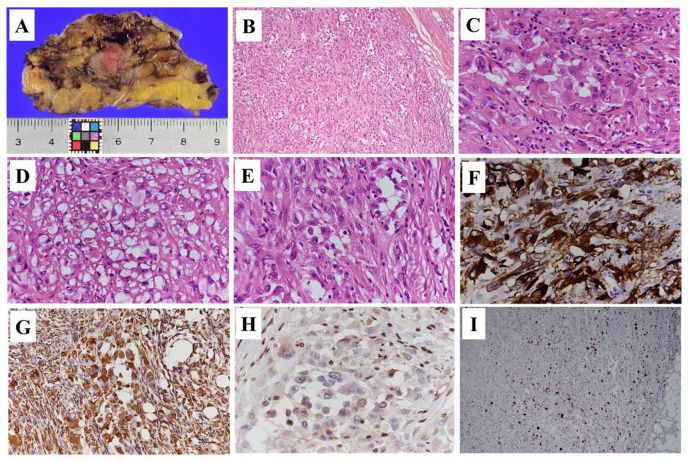
Gross examination of the resected specimen revealed a lobulated and ill-defined, gray-whitish tumor, measuring 18 × 16 × 13 mm in size (**A**). Histopathology revealed that tumor cells exhibited eosinophilic cytoplasm, were polygonal or rounded, cohesive, and arranged in an epithelial-like pattern. Spindle-shaped tumor cells were intermixed, displaying a striform pattern (**B**). Epithelioid cells with peripheral nuclear positioning proliferated in a nodular or nest-like arrangement (**C**), and alongside epithelioid cells, tumor cells exhibiting a polyhedral shape were intermixed (**D**). Rhabdoid cells, with strong eosinophilic cytoplasm containing inclusion-like structures, proliferated in a nodular pattern (**E**). Immunohistochemistry (IHC) was positive for cytokeratin CAM5.2 (**F**), AE1/AE3, epithelial membrane antigen, and vimentin (**G**), with loss of SMARCB1/INI1 expression (**H**). The Ki67 (MIB-1) labeling index was 20% (hot spot) (**I**). These findings confirmed epithelioid sarcoma with negative surgical margins. At 6-month follow-up, there was no evidence of recurrence or metastasis. ES is a rare malignancy, accounting for less than 1% of adult soft tissue sarcomas [1]. A characteristic feature of ES is its broad age range at onset, although it is more commonly observed in individuals aged between 20 and 40 [2]. The male-to-female ratio is approximately 2:1, with a higher prevalence in men. Clinical symptoms commonly include slow-growing, painless, and firm nodules. In contrast, our case presented with a painful lesion. Considering the exceptionally small size and anatomical location of the lesion, inflammatory conditions such as lymphadenitis or inflammatory pseudotumor were suspected. ES is classified into proximal and classic (distal) subtypes based on pathological findings. The classic subtype is reported to occur approximately twice as frequently as the proximal subtype [1]. Characteristics of the proximal subtype typically include truncal location, occurrence in older individuals, deep tissue origin, as well as a tendency to be larger and more aggressive [2]. While the etiology of ES remains unknown, some cases have been reported following trauma [1]. In this case, the lesion presented in the inguinal region with accompanying pain, initially raising suspicion for lymphadenitis or epidermal cysts. However, the differential diagnosis spanned a wide range from benign and malignant tumors to inflammatory conditions. Additionally, differential diagnoses may vary based on gender and require careful evaluation [3]. Considering the presence of pain and differentiation from a female perspective, potential conditions include cystic adenomyosis, desmoid tumor, hydrocele, iliopsoas abscess or bursa, and lymphadenopathy or lymphadenitis. Therefore, careful evaluation using ultrasound, CT, and MRI is necessary to refine the diagnosis as required [3]. In this case, neither the size nor the MRI findings indicated significant suspicion for sarcoma, highlighting the limitations of imaging studies. Therefore, pathological examination via biopsy is recommended when needed. Regarding MRI findings in ES, the imaging characteristics are not particularly distinctive. Based on previous reports, T1-weighted images often show low-signal intensity, while T2-weighted images generally exhibit heterogeneous iso- to mildly high signal intensity [4,5,6]. The median tumor size has been reported to be approximately 5 cm [4,5,6]. Other imaging findings include multilobulated morphology, indistinct margins, and surrounding edema or infiltration patterns; however, these features are also observed in other sarcomas, making differentiation solely through imaging challenging. In contrast, the present case lacked the previously reported characteristics of ES, potentially due to the small tumor size. The lesion appeared as a solitary nodule with well-defined margins and no surrounding edema. PET-CT is considered useful for detecting primary and metastatic lesions of ES, and its SUVmax showed a relatively high average of 6.9 compared to other sarcomas [7]. Pathological findings of proximal ES consist of multinodular distribution and sheets of larger, more atypical cells with variable rhabdoid morphology [8]. On the other hand, IHC findings show little to no differences between the proximal and classic subtypes of ES [8]. Cytokeratin, epithelial membrane antigen (EMA), and vimentin are typically positive; however, it is important to note that the positivity rates for CK7, CK20, and CK5/6 are reported to be low [8]. Loss of SMARCB1/INI1 expression is a novel finding in ES, observed in approximately 90% of cases [8]. The differential diagnosis based on pathological findings is extensive, ranging from benign and inflammatory to malignant conditions. Benign and inflammatory lesions include granuloma annulare, various types of necrotizing granulomas, rheumatoid nodule, benign fibrous histiocytoma, and giant cell tumor of the tendon sheath [8]. Malignant conditions include epithelioid malignant peripheral nerve sheath tumor, extraskeletal myxoid chondrosarcomas, epithelioid angiosarcomas, melanoma, and metastatic carcinomas [8]. Although loss of INI1 expression is characteristic of ES, it has been reported in several tumors, including myoepithelial tumors, malignant rhabdoid tumors, and medullary carcinomas of the kidney [8]. Thus, careful consideration is required when diagnosing based on pathological findings. The main treatment is wide surgical excision; however, postoperative therapies or radiation therapy may be administered for metastatic lesions [3]. Various chemotherapy regimens have been reported, though none have achieved satisfactory outcomes [2]. The prognosis is reported to include a 5-year survival rate of 45–70% and a 10-year survival rate of 45–66% [1]. Prognostic factors include older age, tumor size greater than 5 cm, high histological grade, proximal subtype, and nodal involvement [1,2]. Sarcomas may mimic benign conditions due to small size or indolent growth, highlighting the need for histopathological evaluation and careful follow-up [9].

## Data Availability

The data presented in this study are available from the corresponding author upon reasonable request.

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
