# Peer review of "Proximal Epithelioid Sarcoma Mimicking Inguinal Inflammation"

_diagnostics, 2025, doi:10.3390/diagnostics15172246_

Round 1

Reviewer 1 Report

Comments and Suggestions for Authors

Dear Authors,

This is a well-written "Interesting Images". Please ensure that on line 62, there is a paragraph separation between the figure 3 caption and the next paragraph. Also, it would be beneficial if the patients' treatment could be outlined before the biopsy (antibiotics, etc.) 

Author Response

Dear Reviewers,

Thank you for your thoughtful feedback and helpful suggestions. We greatly appreciate your time and effort in reviewing our manuscript. Below, we outline our responses to each comment, along with the revisions made, which are highlighted in red in the manuscript.

Reviewer1

This is a well-written "Interesting Images". Please ensure that on line 62, there is a paragraph separation between the figure 3 caption and the next paragraph. Also, it would be beneficial if the patients' treatment could be outlined before the biopsy (antibiotics, etc.) .

Response: Thank you for your comments. I revised the sentences accordingly.

       (Line 46, 70)

Reviewer 2 Report

Comments and Suggestions for Authors

Dear Authors,

I have read carefully and with interest your manuscript highlighting the diagnosis and follow-up  of a patient with uncommon site of epithelioid sarcoma. 

The manuscript is overall well written, and imaging provided is quite clear; however, since in the Figure 2 caption it is mentioned that no distant disease sites were detected, it would be useful to add a maximum-intensity projection (MIP) of the PET-CT performed, displaying all the patient's body. Also, specify which radiopharmaceutical was used for PET-CT (presumably 18F-FDG but it is not reported anywhere in the text). 

Moreover, it would be interesting to add some mention about the usefulness of PET-CT in this condition, with reference to existing literature on this topic.  

Best regards. 

Author Response

Dear Reviewers,

Thank you for your thoughtful feedback and helpful suggestions. We greatly appreciate your time and effort in reviewing our manuscript. Below, we outline our responses to each comment, along with the revisions made, which are highlighted in red in the manuscript.

Reviewer2

I have read carefully and with interest your manuscript highlighting the diagnosis and follow-up of a patient with uncommon site of epithelioid sarcoma.

The manuscript is overall well written, and imaging provided is quite clear; however, since in the Figure 2 caption it is mentioned that no distant disease sites were detected, it would be useful to add a maximum-intensity projection (MIP) of the PET-CT performed, displaying all the patient's body. Also, specify which radiopharmaceutical was used for PET-CT (presumably 18F-FDG but it is not reported anywhere in the text).

Moreover, it would be interesting to add some mention about the usefulness of PET-CT in this condition, with reference to existing literature on this topic. 

Response: Thank you for your comments. I revised the sentences accordingly

       (Line 52, 103-105) and added the whole body PET-CT in Figure2B